# Impact of Global Health Scholarship Programs in the Faculty of Medicine at Mbarara University of Science and Technology

Jonans Tusiimire [1,*], Miriam Josephine Nakiwala [1], Brian Turigye [1], Daphine Ansiimire [1], Annet Kembabazi [2], Stephen Asiimwe [2,3] and Joseph Ngonzi [1]

[1] Faculty of Medicine, Mbarara University of Science and Technology (MUST), Mbarara P.O. Box 1410, Uganda
[2] Global Health Collaborative, Mbarara University of Science and Technology (MUST), Mbarara P.O. Box 1410, Uganda
[3] Massachusetts General Hospital Center for Global Health (MGH-CGH), Boston, MA 02114, USA
[*] Correspondence: jonanstusiimire@must.ac.ug; Tel.: +256-774521094

**Abstract:** In recognition of the critical role of residency programs in narrowing healthcare inequalities, Global Health scholarships were introduced at Mbarara University of Science and Technology (MUST) in 2011. Since then, at least 154 postgraduate students in priority programs have benefited. We conducted an online cross-sectional survey to examine how the scholarships and beneficiaries have impacted MUST and the community. Fifty (50) beneficiaries, representing 32.5%, responded, of whom 36 (72%) were alumni. Most respondents were males ($n = 30$; 60%) pursing Master of Medicine ($n = 29$; 58%) or Master of Nursing Science ($n = 20$; 40%) programs. The scholarship schemes included *First Mile* ($n = 29$; 58%), *Kayanja* ($n = 12$; 24%), *Paiko* ($n = 5$; 10%) and *Seed* ($n = 4$; 8%). The majority of the scholarships supported both tuition and research fees ($n = 41$; 82%), the rest being partial. Career advancement was undertaken by eight (16%) of the scholars in the form of fellowships ($n = 3$; 6%), other masters ($n = 3$; 6%) and PhDs ($n = 3$; 6%), with some students having attained a combination of these. All scholars belonged to at least one health professional association. Over 88% ($n = 32$) of the alumni and 28% ($n = 4$) of the students were employed. The majority of those employed were in the public sector (n = 24; 66.7%), mainly the health sector (n = 18; 50%), academia ($n = 14$; 38.9) or both ($n = 4$; 11.1%). There was a high impact on health care provision, undergraduate training and research carried out by the scholars both during training and post-graduation. High levels of career satisfaction, scholarship impact and academic program relevance were reported. The findings provide insights on how low-fund specialty scholarships can have a far-reaching impact on local training, health care and research in low- and middle-income countries.

**Keywords:** alumni satisfaction; tracer study; scholarship; global health collaborative; medical education

## 1. Introduction

The Global Health Collaborative (GHC) is a collaboration primarily between Mbarara University of Science and Technology (MUST) and Mbarara Regional Referral Hospital (MRRH) in Uganda with Massachusetts General Hospital (MGH) in Boston, USA, and other global health entities. Over the last 15 years, GHC has pursued its mission to build bilateral institutional capacity at MUST, MRRH and MGH through research, innovation, education, clinical care and responsive program implementation in Uganda and the United States. As part of this mission, GHC has supported various initiatives at MUST and MRRH in teaching, research, medical innovation, clinical care and community service (https://globalhealth.massgeneral.org/ourwork-items/global-health-collaborative/ accessed on 15 September 2022). In recognition of the crucial role of local clinical residency training programs in narrowing the global healthcare gap in low- and middle-income countries (LMICs) such as Uganda [1], particularly inequalities in specialized care [2,3], GHC has been offering scholarships to postgraduate students in the MUST Faculty of Medicine since the 2011/2012 academic year. The scholarships were intended for capacity building

in priority medical and surgical specialties, so as to improve the quality and safety of health care [4], while boosting MUST/MRRH's capacity as a training hub for modern-day specialists [1,5]. These scholarships have been offered through various schemes, including Paiko, Sullivan, SEED, Kayanja and First Mile, the latter being the latest and most prominent of the scholarship schemes. Scholarships under Paiko were the first to be awarded during the 2011/2012 academic year. Later, the Sullivan, SEED and Kayanja schemes were introduced between the 2014/2015 and 2015/2016 academic years. These schemes offered scholarships ranging from one to nine per year, with the majority being under Paiko.

Notably, during the 2018/2019 academic year, the *First Mile* program was also introduced under the same collaboration (https://www.must.ac.ug/first-mile-scholarship-opportunities-2019-2020/ accessed on 15 September 2022), which significantly increased the number of scholars. Unlike the previous schemes, First Mile is a multipronged program which supports many academic, research and community-based initiatives at MUST and MRRH, particularly in the departments of Nursing and Community Health. This philanthropy-supported scheme, funded by Hansjörg Wyss through MGH, was intended "to power the Academic Medical Center to deliver healthcare in the community in Uganda". The program was driven by the belief that everyone everywhere has a right to good health. The First Mile program aims to (1) empower nursing leadership to develop and implement innovative models of care; (2) establish MUST as a premier academic medical center focused on community-based healthcare delivery, research and innovation for East Africa; (3) equip the MRRH and affiliated community health facilities to better understand and meet the burden of disease; and (4) leverage technology innovation and co-creation to support patients, community health workers and nurses (https://www.must.ac.ug/first-mile-scholarship-opportunity-2022-2023-call-for-applications/ accessed on 15 September 2022).

By and large, the First Mile program aims at creating a strong and sustainable bond between the MUST Faculty of Medicine/MRRH and the neighboring community for better health service provision. This is in line with the MUST Faculty of Medicine's strategic goal to consolidate and enhance university outreach and community engagement programs in accordance with the Community-Based Education, Research and Services (COBERS) philosophy for training health care professionals. Under First Mile, specially trained doctors, nurses, and Village Health Team (VHT) workers are sent deep into communities and schools to train and sensitize people to the prevention and treatment of a variety of common diseases.

The postgraduate programs in the MUST Faculty of Medicine are competency-based, emphasizing the acquisition of appropriate knowledge, skills (both hands-on and soft) and attitudes by the learners to be able to deliver effectively in the workplace. The programs are delivered through a blend of didactic courses, clinical sessions, clerkships and self-directed learning [6]. All the GHC scholarships are competitive and merit-based, and they target talented Ugandan trainees already admitted into specific, priority postgraduate programs in the MUST Faculty of Medicine who are not on any other scholarship or receiving financial support from employers, organizations or the Ministry of Health. The awards cover tuition fees, research support or both for the entire duration of the program, which is either 2 years (MSc/MNS programs) or 3 years (MMed programs). The programs supported by these scholarships vary each year depending on the needs and gaps identified by the Faculty of Medicine. In turn, the scholars are required to study full-time on their programs, file progress reports at the end of each semester, participate in the program's outreach and community activities, and generally participate in any other activities integral to their programs' curricula for the duration of the scholarship. Until now, there have not been any previous formal follow-ups to document the scholars' accomplishments.

Alumni tracer studies are surveys of a homogenous group of trainees that are conducted some period after graduation or at the end of training to assess their course of study, the transition to work, employment status, career and application of learned competencies [7–9]. The majority of impact evaluation studies on scholarships and capacity-building

programs tend to focus on individual recipient outputs and outcomes [10]. However, a more acceptable approach is to look at impact in a holistic way to gain an understanding of the outcomes of key interventions on the beneficiaries, communities and policy. Routinely, these studies rely on the use of mixed-methods approaches, longitudinal designs and alumni tracking to gain a deep understanding of the various program elements needed to bring about change at the individual, community and societal levels [11].

Since the inception of these scholarships, many scholars have successfully enrolled, trained and graduated in these programs and subsequently served their communities in different capacities. However, there has not been any systematic study to trace these alumni post-graduation. As a result, little is known regarding the career progress and impact of scholars. Given the amount of time since the first scholars were admitted and the number of scholars that have been sponsored since, it was apparent that a study of this nature was needed. Therefore, this study aimed to assess the impact these scholarships offered to the MUST Faculty of Medicine in the provision of evidence-based health care, innovations, transformational leadership and promotion of socioeconomic development at the community level. In turn, this would inform the key stakeholders (the university and scholarship management) on whether the key objectives of the scholarship were being met and the impact the scholarship was having on the health and economic development of Ugandan communities.

## 2. Materials and Methods

Study design: This was an online cross-sectional survey conducted from 29 November to 19 December 2022 among the GHC-sponsored scholars and alumni of residency training programs of the MUST Faculty of Medicine. This study utilized a pre-validated tool containing both quantitative and qualitative questions.

Setting: The Faculty of Medicine is the oldest and pioneer faculty of MUST, a public university established in October 1989. MUST is affiliated to MRRH, a public referral and teaching hospital that serves about 5 million people mainly from southwestern Uganda [6]. The faculty's vision, mission and philosophy are geared toward attaining excellence in health sciences education, research, innovation and community service. Over the past 33 years, the faculty has grown from just one undergraduate program of Bachelor of Medicine and Bachelor of Surgery (MBChB) to six undergraduate direct [12], three undergraduate completion, one diploma and twenty-five master [6] and PhD programs in various specialties (Table 1). Current undergraduate degree programs include MBChB, nursing science (direct and completion), pharmacy, pharmaceutical sciences, physiotherapy and medical laboratory sciences (direct and completion). Postgraduate programs include masters of medicine (various specialties), pharmacy (clinical pharmacy), nursing science (various specialties), medical laboratory science (various specialties) and public health (including research ethics) and master of science in anatomy, biochemistry, medical microbiology, pharmacology, physiology, pharmacognosy and natural medicine science and pharmaceutical analysis. Additionally, MUST runs a diploma in child and adolescent mental health in collaboration with Butabika National Referral Hospital and a diploma in emergency medicine in collaboration with Masaka Regional Referral Hospital.

Study population: The study population included alumni and current scholars of the MUST Faculty of Medicine who received any form of scholarship from GHC under any of the existing or previous schemes, such as Paiko, Sullivan, SEED, Kayanja and First Mile. Only scholars and alumni who consented to and responded to the survey were included.

Sample size and selection criteria: Since this study focused on a fairly small population, a census approach was used. As inclusion criteria, all current or past GHC scholarship awardees who were alive and reachable at the time of this study and consented to participate were considered. A list of all the scholars containing their respective email addresses and phone numbers was compiled from past records in liaison with the GHC Secretariat, and any gaps in the contacts were addressed by contacting members of different cohorts.

**Table 1.** Specialties for postgraduate programs in the Faculty of Medicine at MUST.

| Program | Specialties | Duration (Years) |
|---|---|---|
| Master of Medicine | Anesthesia, dermatology, ENT, emergency medicine, general surgery, internal medicine, obstetrics and gynecology, ophthalmology, pediatrics and child health, pathology, psychiatry, radiology | 3 |
| Master of Nursing Science | Critical care nursing, community midwifery and reproductive health, pediatric clinical nursing, mental health nursing | 2 |
| Master of Public Health | Public health, research ethics | 2 |
| Master of Pharmacy | Clinical pharmacy | 2 |
| Master of Medical Laboratory Science | Clinical chemistry, microbiology, histopathology | 2 |
| Master of Science | Biochemistry, anatomy, physiology, medical microbiology, pharmacology, pharmaceutical analysis, pharmacognosy, natural medicine science | 2 |

Data collection: Data were collected using an online questionnaire sent to the participants in an email link. The questionnaire was developed initially in paper format by a team of faculty from the MUST Faculty of Medicine and transferred onto the KoboToolbox platform (version: v2021.2.4). A link to the questionnaire (see: Supplementary Materials) was emailed to a senior researcher with experience in survey design and qualitative research for validation. The validated questionnaire was then pretested on one current GHC scholar and one recent graduate to confirm the clarity of the questions. The final 29-item questionnaire contained six sections on (a) background information about the scholar and scholarship, (b) additional trainings and professional affiliations post-scholarship, (c) the scholar's impact on the university during the scholarship, (d) current employment-related information, (e) impacts of the scholarship and (f) the scholar's satisfaction. The qualitative data were in the form of written descriptions in response to three open-ended questions addressed to the scholars. The first question ("In what way(s) did you make (or are you making) significant positive change to the university?") inquired about the scholars' positive impacts on the university, particularly the medical school community (including teaching hospital), during the scholarship. The second ("Briefly describe your major career achievements since graduation") and third ("State your major contributions to healthcare and community development since obtaining the scholarship") questions referred to post-scholarship career achievements and contributions to healthcare and community development, respectively. A Likert scale was used to explore the data on scholarship, program relevance and level of career satisfaction, using (1) very high, (2) high, (3) somewhat, (4) limited and (5) not at all. Where applicable, the interview questions were designed in such a way that they took into account whether the scholars had completed their programs.

Data management and analysis: All data collected were automatically synchronized onto a password-protected Kobo Toolbox platform, which hosted the online tool. This permitted real-time data capture and entry, minimized errors at entry and eased data cleaning. At the end of data collection, the dataset was downloaded into a Microsoft Excel (version: 2016) spreadsheet for cleaning. Data cleaning involved checking for accuracy, completeness and consistency of data. Quantitative data were then exported to SPSS version 20 software for statistical analysis, and the data summarized in tables as frequencies and percentages. The Likert scale responses were summarized as bar charts using Microsoft Excel [13].

For qualitative data, the responses to each of these questions were read and reread by two investigators who worked together to highlight and label blocks of texts, took notes and made sense of the texts. Manifest content analysis [14] was employed. This analysis offers a simple and objective approach which focuses on easily observable targets within textual data without the need to discern intent or identify deeper meanings [14]. This was deemed appropriate for our study based on the expectation and assumption that the

scholars voluntarily provided objective information about their experiences, career paths and contributions, and so there was no reason not to take them at face value [14,15].The relevant pieces of information identified from the texts were carefully coded. The codes with related meanings were then grouped into categories, then related categories described with a specific theme. Further analysis of these themes was carried out to determine how they were interrelated.

### 3. Results

Over the past 10 years, the Faculty of Medicine at MUST has hosted a total of 154 students on GHC scholarships in 22 postgraduate programs. These include 81 (52.6%) Master of Medicine (MMed) programs, 57 (37.0%) Master of Nursing Science (MNS) programs, 7 (4.5%) Master of Science (MSc) programs, 6 (3.9%) Master of Medical Laboratory Science (MMLS) programs and 3 (1.9%) Master of Public Health (MPH) programs (Table 2). The MSc programs that have thus far participated in the scholarships include biochemistry (four scholars), microbiology (two scholars) and physiology (one scholar). For MMLS, the subspecialties pursued by scholars thus far include parasitology (two scholars), histopathology (one scholar), medicine (one scholar), and laboratory diagnostics (one scholar).

**Table 2.** Programs that have received global health scholars in the MUST Faculty of Medicine.

| SN | Academic Program | Program Category | N (%) |
|----|------------------|------------------|-------|
| 1 | Critical care nursing | MNS | 57 (37.0) |
| 2 | Pediatrics and child health | MMED | 16 (10.4) |
| 3 | Obstetrics and gynecology | MMED | 11 (7.1) |
| 4 | Psychiatry | MMED | 11 (7.1) |
| 5 | Anesthesia | MMED | 8 (5.2) |
| 6 | Internal medicine | MMED | 8 (5.2) |
| 7 | Surgery | MMED | 7 (4.5) |
| 8 | Medical laboratory science | MMLS | 6 (3.9) |
| 9 | Pathology | MMED | 6 (3.9) |
| 10 | Radiology | MMED | 6 (3.9) |
| 11 | Biochemistry | MSc | 4 (2.6) |
| 12 | Ear, nose and throat | MMED | 3 (1.9) |
| 13 | Ophthalmology | MMED | 3 (1.9) |
| 14 | Master of Public Health | MPH | 3 (1.9) |
| 15 | Microbiology | MSc | 2 (1.3) |
| 16 | Dermatology | MMED | 1 (0.6) |
| 17 | Emergency medicine | MMED | 1 (0.6) |
| 18 | Physiology | MSc | 1 (0.6) |
| | Total | | 154 |

*Demographic information of the scholars:* Out of the 154 scholars contacted through the online questionnaire, 50 (32.5%) responded. Of these, 36 (72%) were alumni, and the rest were current students. The majority were male ($n = 30$; 60%) and graduates of MMED ($n = 29$; 58%) or MNS ($n = 20$; 40%) programs, and only one was from the MMLS program. The majority of the MMED respondents were from pediatrics and child health ($n = 5$), obstetrics and gynecology ($n = 5$) and anesthesia ($n = 4$). Others were from ophthalmology ($n = 3$), ENT ($n = 2$), general surgery ($n = 2$), radiology ($n = 3$), psychiatry ($n = 1$), dermatology ($n = 1$), histopathology ($n = 1$), internal medicine ($n = 1$) and pathology ($n = 1$). Table 3 shows the detailed demographics of the participants.

**Table 3.** Summary of the demographic information for the 50 participants involved in this study.

| Variable | Values | N (%) |
|---|---|---|
| Gender | Male | 30 (60) |
| | Female | 20 (40) |
| Field of study | Nursing | 20 (40) |
| | Medicine | 29 (58) |
| | MLS | 1 (2) |
| Program of study | MNS | 20 (40) |
| | MMED | 29 (58) |
| | MMLS | 1 (2) |
| Specific program | Anesthesia | 4 (8) |
| | Pediatrics and child health | 5 (10) |
| | Psychiatry | 1 (2) |
| | General surgery | 2 (4) |
| | Other | 5 (10) |
| | Critical care nursing | 20 (40) |
| | Dermatology | 1 (2) |
| | Ear, nose and throat | 2 (4) |
| | Histopathology | 1 (2) |
| | Internal medicine | 1 (2) |
| | Obstetrics and gynecology | 4 (8) |
| | Ophthalmology | 3 (6) |
| | Pathology | 1 (2) |
| Year of scholarship award | 2012 | 2 (4) |
| | 2013 | 1 (2) |
| | 2014 | 3 (6) |
| | 2015 | 4 (8) |
| | 2016 | 7 (14) |
| | 2018 | 16 (32) |
| | 2019 | 7 (14) |
| | 2020 | 4 (8) |
| | 2021 | 2 (4) |
| | 2022 | 4 (8) |
| Scholarship scheme | First Mile | 29 (58) |
| | Kayanja | 12 (24) |
| | Paiko | 5 (10) |
| | SEED | 4 (8) |
| Form of support | Tuition and research | 41 (82) |
| | Tuition only | 2 (4) |
| | Part tuition | 4 (8) |
| | Part tuition and research | 2 (4) |
| | Research only | 1 (2) |

**Table 3.** *Cont.*

| Variable | Values | N (%) |
|---|---|---|
| Completion status | Yes | 36 (72) |
| | No | 14 (28) |
| Year of completion | 2015 | 1 (2) |
| | 2016 | 5 (10) |
| | 2017 | 4 (4) |
| | 2018 | 7 (14) |
| | 2019 | 8 (16) |
| | 2020 | 1 (2) |
| | 2021 | 6 (12) |
| | 2022 | 4 (8) |
| Additional scholar training post master's scholarship (n = 36) | Yes | 8 (16) |
| | No | 28 (56) |
| If "yes" additional training post master's scholarship | Fellowship | 3 (6) |
| | Masters | 2 (4) |
| | Masters and PhD | 1 (2) |
| | PhD | 2 (4) |
| Additional training during current scholarship | Yes | 2 (4) |
| | No | 15 (30) |
| If "yes" to the above | Responsible conduct of research | 1 (2) |
| | Responsible conduct of research and data analysis | 1 (2) |
| Highest academic qualification | Bachelors | 17 (34) |
| | Masters | 33 (66) |
| Professional memberships | UMDPC | 28 (56) |
| | UNMC | 19 (38) |
| | Other | 3 (6) |

The respondents had received their scholarship awards from as early as 2012 to as recently as 2022, but the majority were 2018 awardees (*n* = 16; 32%). Only scholarship awardees of the 2017 cohort were not represented among the respondents.

Most of the respondents were First Mile scholars (*n* = 29, 58%), followed by Kayanja (*n* = 12, 24%), Paiko (*n* = 5, 10%) and SEED (*n* = 4, 8%) scholars. The majority of the scholarships had supported both tuition and research fees (82%), and the rest were tuition only (4%), part tuition (8%), part tuition and research (4%) or research only (2%). Seventy-two percent (*n* = 36) of the scholars had completed their programs at the time of this study, while the rest (*n* = 24) were still in the programs.

Most of those who had completed their programs had done so between 2015 and 2022, and 16% of them had gone on to pursue additional trainings in the form of fellowships (6%), other masters (6%) and PhDs (4%), although none of them had completed the latter. Two of the current scholars reported having had additional training in the responsible conduct of research and quantitative data analysis. Thus, the highest academic qualification for the respondents was a master's degree in the case of those who had completed the scholarship or a bachelor's degree for those who were still in their postgraduate programs at the time of this study.

On average, each scholar belonged to at least one registered professional body: the Uganda Medical and Dental Practitioners Council (UMDPC) (56%), Uganda Nurses and Midwives Council (UNMC) (38%) or other (6%). The "other" were the Uganda Fertility

Society (UFS), Allied Health Professionals and Critical Care Nursing Association of Uganda (CCNAU), each of which had one participant.

Current employment status of the scholars: Among the alumni, over 88 percent (*n* = 32) were already employed; however, just 28 percent (*n* = 4) of the students had formal jobs (Figure 1). The majority of those employed mainly worked in the public sector (66.7%), and the rest worked in the private sector (33.3%). The main sectors of employment were the health sector (50%) and academia (38.9), with 11.1% of the scholars being employed in both sectors. All those employed were based in Uganda (Table 4).

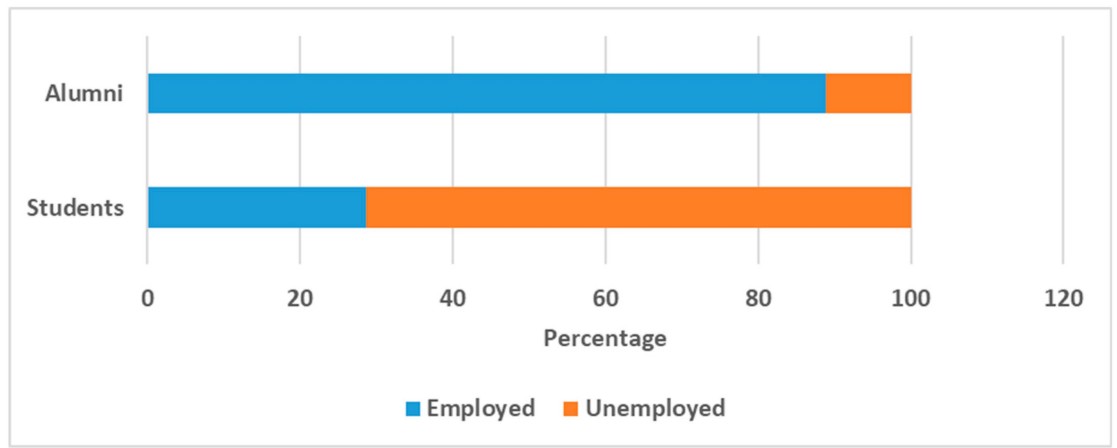

**Figure 1.** Employment status of the GHC scholars, both current students and alumni.

**Table 4.** Summary of the employment details of the scholars (both alumni and students).

| Variable | Values | N (%) |
|---|---|---|
| Employment status | Employed | 36 (72.0) |
| | Unemployed | 14 (28.0) |
| Form of employment | Public | 24 (66.7) |
| | Private | 12 (33.3) |
| Employment sector | Academia | 14 (38.9) |
| | Health sector | 18 (50.0) |
| | Both health sector and academia | 4 (11.1) |
| Employment country | Uganda | 36 (100) |

Relevance of graduate programs to market needs: The scholars reported that they deemed the programs sponsored by the GHC scholarship schemes to be relevant to market needs, with 96% of the respondents reporting that the relevance to the market was very high or high. Only two of the scholars (4%) deemed the sponsored programs to be just somewhat relevant to market needs. Figure 2 illustrates the participants' responses regarding the relevance of the sponsored programs and scholarships to market needs.

Relevance of scholarship to scholar's career: Regarding the relevance of the scholarships to the scholars' needs, all but one scholar reported that the scholarships were very highly or highly relevant, representing 80% and 18% of the respondents, respectively (Figure 2).

Level of career satisfaction among alumni: Approximately 86 percent of the alumni reported that their level of satisfaction with their careers was very high (50%) or high (36%). Only 14% of the scholars reported being somewhat satisfied, but there were no scholars who reported limited satisfaction or total dissatisfaction with their careers (Figure 2).

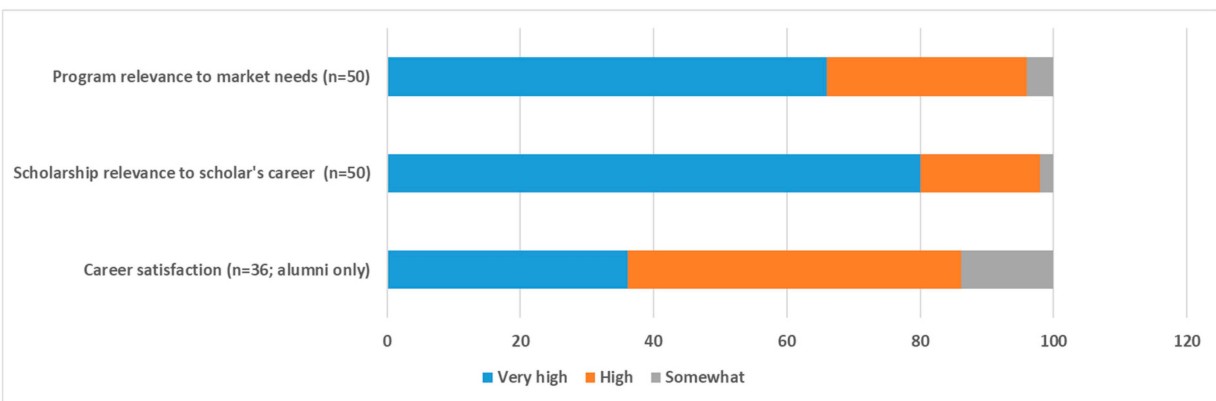

**Figure 2.** Scholar perceived career satisfaction, relevance of graduate program and scholarship.

Perceived scholar impact while on campus: Over 88 percent of the scholars (alumni) perceived themselves to have made a significant positive impact while on campus, but the rest were not sure. Among the current scholars, self-perception of making a significant impact on campus was 71.4%, with the rest not sure whether they were making any significant impact or not. None of the scholars perceived their impact while on campus to have been certainly insignificant (Table 5).

**Table 5.** Scholars' self-perception of having a significant impact while on campus.

| Scholar Category | N (%) | | |
|---|---|---|---|
| | Yes | No | Not Sure |
| Alumni | 32 (88.9) | 0 (0.0) | 4 (11.1) |
| Students | 10 (71.4) | 0 (0.0) | 4 (28.6) |

**Contributions of the scholars while on campus:** Based on the qualitative data from the open-ended question, "In what way(s) did you make (or are you making) significant positive change to the university?", the scholars' contributions to the university and the teaching hospital while on campus during their training were categorized into four major themes: provision of patient care; leadership, advocacy and coordination; conducting research; and training and mentorship of undergraduate students.

Provision of patient care: Most of the GHC scholars reported having provided quality and specialized healthcare services to outpatients during clinics and to inpatients during ward rounds at MRRH. Additionally, they provided patient care services to outreach hospitals and health centers within rural southwestern Uganda during community placements. Even as residents, some of these scholars would run scans in the radiology department, take care of critically ill patients in the ICU (particularly MNC Critical Care students) and attend to those requiring emergency care in the hospital's emergency department. The following quotes illustrate the range of clinical care that the scholars provided:

*"I also assisted in running the CT scan examinations at Mbarara Regional Referral Hospital following the re-establishment of procedures".*
*(Scholar P021, MMED Radiology)*

*"I contributed to clinical care in the ICU, theatre, emergency department and research areas".*
*(Scholar P022, MMED Anesthesia)*

*"From my critical care training, I was able to provide quality care to emergency and critically ill patients to enhance their recovery".*
*(Scholar P020, MNS Critical Care)*

Alumni of internal medicine had the opportunity to support cancer care during their residency training. One respondent notes that during their time as a resident at MRRH, the oncology unit was under the care of the department of internal medicine. This gave the residents an opportunity to train in oncology care while also contributing to the management of cancer patients.

> *"I contributed to the clinical care of patients with cancer at the hospital [oncology unit], which was under the supervision of the department of [internal] medicine".*
>
> *(Scholar P047, MMED Internal Medicine)*

It was notable that the range of contributions made by the scholars was mainly in the various departments of the teaching hospital, but in some cases, the impact was in the placement sites where these scholars practiced, particularly during the recess periods of their training. During such placements, the scholars had the opportunity to perform specialized interventions, besides training undergraduates, as this participant notes:

> *"I performed several specialized surgeries during my placement at Kisoro district hospital. Offered bedside teaching to undergraduate medical students".*
>
> *(Scholar P032, MMED Surgery)*

It was quite apparent that the training at MUST had improved the scholars' skills to the extent that the services they offered were perceived to be of higher quality. The residents continued to offer this high-quality clinical care even after completion of their residency programs. The alumni expected that the improved care offered to patients would add to the good reputation of the university. This probably implied that alumni were happy to remain associated with their *alma mater* and that the public would easily associate the quality of their services with the university that had nurtured them. They sought to be good ambassadors.

> *"I have since utilized the training to serve people in Kabale, a rural area in Western Uganda".*
>
> *(Scholar P005, MMED Obs/Gyn)*

> *"My skills in clinical care were improved, and so I was able to care for patients in a way that adds to the already good reputation of the university".*
>
> *(Scholar P031, MNS Critical care)*

Leadership, advocacy and coordination: Some of the scholars took on the mantle of leadership and advocacy, which they effectively utilized to ensure quality in service provision, influence and advocate for positive changes in training, health service provision and program coordination and inspire students to advance their knowledge and skills through specialized training.

At the time of receiving the scholarship, some of the scholars were already part of the faculty, retained as teaching assistants after their graduation from the undergraduate programs to support teaching in some nascent departments. Frequently, these young academics, now residents on a reputable scholarship, became role models for undergraduate students who would be inspired to take on careers similar to their teachers', leading to increased enrollment in residency programs. These scholars went on to have wider reaching impacts at the departmental level through leadership, advocacy and lobbying for increased support from university and hospital management. The following quote from a scholar in the radiology department clearly illustrates this point:

> *"I have remained as a teaching staff member in the Faculty of Medicine, where I have participated in the establishment of the department and strengthened radiology teaching for both undergraduates and postgraduates in addition to providing clinical care to patients. To date, there is overwhelming interest from students who want to pursue radiology. Through progressive leadership as head of department, the hospital has invested significantly in equipment to further improve learning and patient care. The student population pursuing Radiology has improved from 1 during my study period to 20 to date".*
>
> *(Scholar P017, MMED Radiology)*

Some of the scholars were appointed as undergraduate class coordinators in their departments. As class coordinators, they were instrumental in advocating for better welfare and support to the learners while mediating residents' issues with the lecturers, heads of departments, offices of the dean and academic registrars. In this way, the class coordinators were critical in determining how undergraduate training was organized, thus leading to an effective learning process. In addition to these roles, such class coordinators provided linkages with other universities to allow sharing of knowledge and experiences, thus raising the profile of the program and university to the outside world. This quote illustrates this role further:

*"I was involved in the practical demonstration for undergraduate students, participated in examination setting for undergraduates and advocacy for fairness and support of the learners. [I] advocated for collaboration with other universities for elective training and sharing of experience and knowledge for increased marketability of students during recess semesters or as may be planned".*

*(Scholar P044, MMED Pathology)*

*"I participated in teaching critical care, and currently, I coordinate the critical care program. In addition, I have published a number of papers that have increased the visibility of the university at large".*

*(Scholar P027, MNS Critical Care)*

*"... I was the chairperson quality assurance and helped to improve quality".*

*(Scholar P041, MMED Obs/Gyn)*

Conducting research: A number of scholars considered research to be one of their key contributions made while on campus. As part of their postgraduate training, the scholars trained and supervised undergraduates as they conducted their own research. Some of this research was published in the form of peer-reviewed articles in journals, which led to increased visibility of the university. In addition to adding to the knowledge base, the scholars reported that some of their research was able to bring about changes in policy as well as inform clinical practice. These quotes clearly summarize these points:

*"I have published a number of papers that have increased the visibility of the university at large".*

*(Scholar P027, MNS Critical Care)*

*"I instilled in students a positive attitude toward nursing and patients and conducted research that was able to bring about policy changes".*

*(Scholar P049, MNS)*

*"My research was able to inform clinical practice".*

*(Scholar P010, MNS Critical Care)*

It was notable that while the majority of research themes centered on works by the scholars themselves, some of the respondents reported that they had trained others to do research. The latter was undertaken as part of program training in research for undergraduate or postgraduate students in which the scholars constituted the team of trainers and/or supervisors. This is illustrated in the following quotes:

*"I [gave] back to the university by teaching both undergraduate and postgraduate students as well as conducting research..."*

*(Scholar P006, MMED Obs/Gyn)*

*"[I was involved in] teaching undergraduate students, clinical work and conducting clinical research".*

*(Scholar P039, MMED Psychiatry)*

*"[I was involved in] teaching undergraduates and undergraduate research supervision".*

*(Scholar P024, MNS Critical Care)*

Training and mentorship: All scholars trained and mentored undergraduate students in the classrooms and/or during clinical rotations, mainly in their respective specialties, but sometimes in other departments when need arose. This training involved teaching and setting examinations and marking them. It is apparent that these scholars were passionate in training their successors, and some took the initiative to practically demonstrate abstract concepts to students and utilized the platform to encourage and inspire trainees in their career choices.

*"As a resident in the department of Radiology, which was understaffed during my residency, I took part in teaching undergraduates (both in class and clinical rotations) and fellow residents in other disciplines (e.g., obstetrics and gynecology)".*

*(Scholar P021, MMED Radiology)*

*"[Besides] participation in clinical care for the patients, [I took] part in clinical teaching of undergraduate students. I was also the postgraduate leader in my department and led to many positive changes, especially in how our work/training was organized".*

*(Scholar P004, MMED Pediatrics)*

*"I participated in teaching and examining the undergraduate students".*

*(Scholar P037, MNS Critical Care)*

**Scholarship impacts on the alumni:** We asked the alumni about their perceived impacts of the scholarship on their current careers. The questions reflected on the extent to which the scholars agreed or disagreed with four specific statements centered on scholarship-enabled attainment of essential skills and competencies for optimal healthcare delivery. The statements encompassed themes on leadership in healthcare innovation, community-based healthcare delivery, community-based research to address disease burden and leveraging healthcare technologies for better patient care. Box 1 briefly describes these themes.

**Box 1.** Scholar skills and competency sets impacted by the scholarship program

> 1. Enhanced leadership skills: The extent to which the scholarship enhanced the scholar's leadership skills and ability to develop and implement innovative health care systems and models of patient care.
> 2. Community-based healthcare: The extent to which the scholarship increased the scholar's competences (knowledge, skills and attitude) in community-based healthcare delivery.
> 3. Community-based research: The extent to which the scholarship equipped the scholar with the skills to better understand and address the burden of disease through community-based research.
> 4. Leveraging healthcare technologies: The extent to which the scholarship equipped the scholar with the ability to leverage existing technologies to innovate and create new solutions for patient management and care.

Enhanced skills in leadership for innovative healthcare: The alumni agreed that the scholarship had a positive impact on their leadership skills to drive innovations in healthcare systems and patient care. Of the thirty-six alumni who responded, over 91% either strongly agreed (n = 13) or agreed (n = 20) that the scholarship had enhanced their leadership skills and ability to develop and implement innovative health care. Conversely, one scholar (2.8%) strongly disagreed with the statement, while two of them (5.6%) neither agreed nor disagreed (Figure 3).

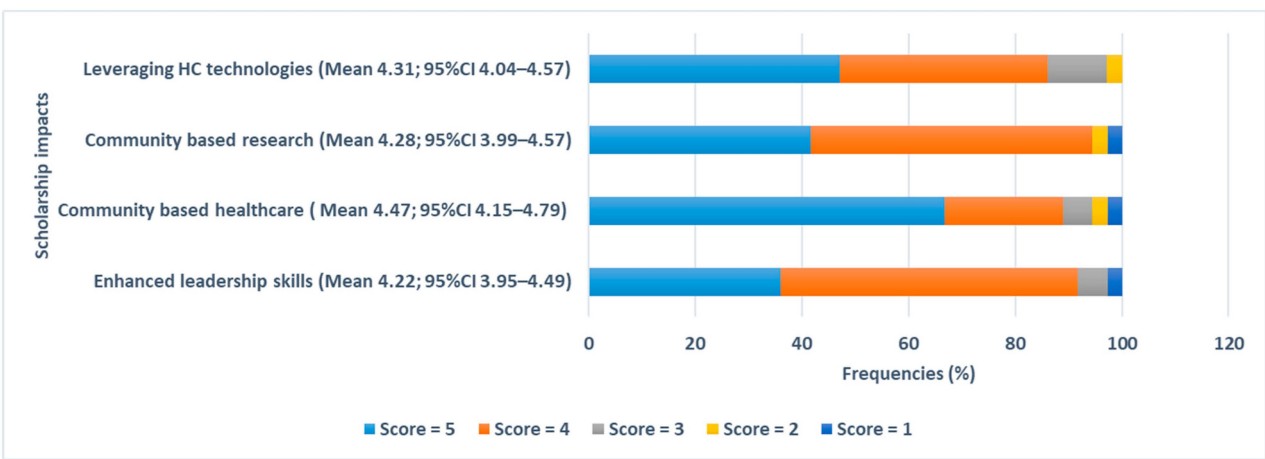

**Figure 3.** Scholar perspectives on the actual scholarship impacts on their careers. Scores are as follows: 5 (strongly agree), 4 (agree), 3 (neutral), 2 (disagree) and 1 (strongly disagree).

Increased competence in community-based healthcare: Over 66 percent (*n* = 24) of the alumni strongly agreed and 22.2% (*n* = 8) agreed that the scholarship had increased their competences (knowledge, skills and attitude) in community-based healthcare delivery. Conversely, 5.6% of the scholars either strongly disagreed (*n* = 1) or disagreed (*n* = 1). Two of the participants (5.6%) neither agreed nor disagreed with the statement (Figure 3).

Understanding and addressing disease burden through community-based research: Almost 42 percent (*n* = 15) of the alumni strongly agreed and 52.8% (*n* = 19) agreed that the scholarship had equipped them with skills to better understand and address the burden of disease through community-based research. Conversely, 5.6% of the scholars either strongly disagreed (*n* = 1) or disagreed (*n* = 1) with the statement (Figure 3).

Leveraging existing technologies to innovate and create new solutions for healthcare: Over eighty-six percent of the alumni either strongly agreed (*n* = 17) or agreed (*n* = 14) that the scholarship enhanced their ability to leverage existing technologies to innovate and create new solutions for patient management and care. Conversely, 2.8% (*n* = 1) of the scholars disagreed. Four participants (11.1%) neither agreed nor disagreed with the statement (Figure 3).

**Anticipated scholarship impacts on the students:** As with the alumni, we asked the students (current scholars) about their expected impacts of the scholarship on their future careers. The same questions, statements and themes were used as with the alumni (see Box 1), only with slight modification in the phrasing to future tense since the impacts were being anticipated. The responses are summarized in Figure 4.

Enhanced skills in leadership for innovative healthcare: The current scholars believed that the scholarship would have a significant impact on their leadership skills to drive innovations in healthcare systems and patient care. Of the seventeen students, 47 percent (*n* = 8) strongly agreed and 23.5% (*n* = 4) agreed that the scholarship enhanced their leadership skills and ability to develop and implement innovative health care. Conversely, those who disagreed or strongly disagreed were 5.9% (*n* = 1) and 23.5% (*n* = 4), respectively.

Increased competence in community-based healthcare: The same trend as above was also observed among the current scholars concerning the anticipated impact of the scholarship on their competences (knowledge, skills and attitude) in community-based healthcare. A total of 47 percent (*n* = 8) strongly agreed and 23.5% (*n* = 4) agreed that the scholarship increased their competences in community-based healthcare. Conversely, those who disagreed or strongly disagreed were 5.9% (*n* = 1) and 23.5% (*n* = 4), respectively.

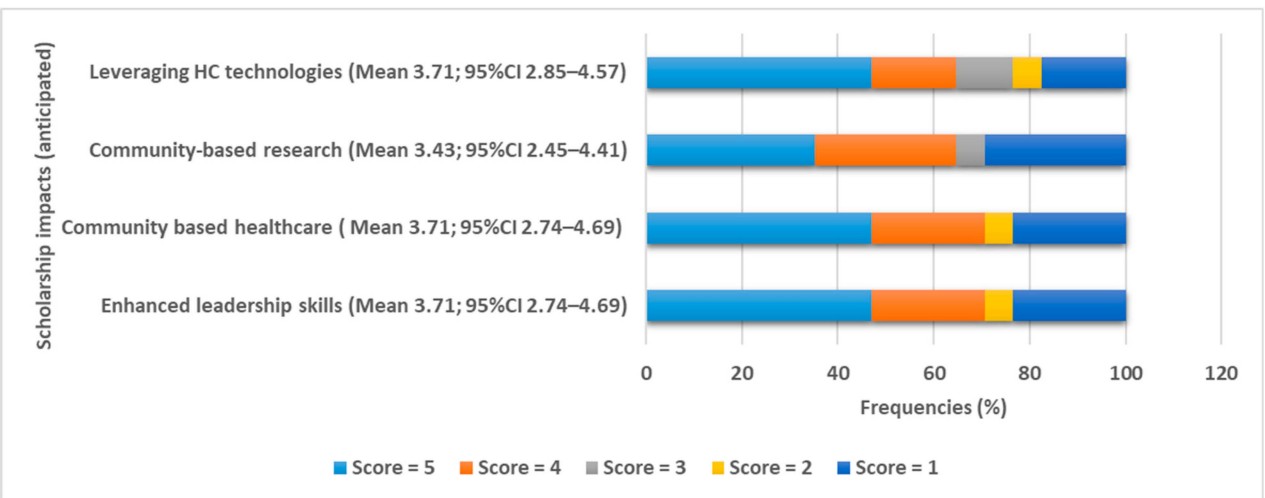

**Figure 4.** Scholar perspectives on the anticipated scholarship impacts on their careers. Scores are as follows: 5 (strongly agree), 4 (agree), 3 (neutral), 2 (disagree) and 1 (strongly disagree).

Understanding and addressing disease burden through community-based research: Over seventy percent of the current scholars strongly agreed (*n* = 6) or agreed (*n* = 5) that the scholarship was equipping them with skills to better understand and address the burden of disease through community-based research. Conversely, 29.4% (*n* = 5) strongly disagreed with this statement. Only one scholar neither agreed nor disagreed.

Leveraging existing technologies to innovate and create new solutions for healthcare: Almost sixty-five percent of the scholars either strongly agreed (*n* = 8) or agreed (*n* = 3) that the scholarship enhanced their ability to leverage existing technologies to innovate and create new solutions for patient management and care. Conversely, 23.5% of the scholars either strongly disagreed (*n* = 3) or disagreed (*n* = 1) with the statement. Two of the scholars neither agreed nor disagreed.

### 3.1. Career Achievements by the Alumni

Based on the qualitative data from the responses to the open-ended question, "Briefly describe your major career achievements since graduation", the scholars' career achievements were categorized into five major themes: academia, research, innovation and pioneering change, professional advancement and leadership.

Achievements in academia: Following their graduation, some of the scholars had sought to pursue their interests in academic scholarship. These scholars were either planning, currently pursuing or had completed another master's degree, a fellowship or PhD program in health sciences. Obviously, these scholars had been retained at MUST or were employed in other medical schools as faculty members, some even earning promotions during the post-scholarship period.

> *"I have done a fellowship in Alzheimer's disease and related dementias, published 10 papers in reputable journals and I am currently undertaking a fellowship in children's palliative care".*
>
> *(Scholar P027, MNS Critical Care)*

> *"I enrolled for a PhD in health sciences (taught; epidemiology and biostatistics). I finished course works and examinations and now writing my concept".*
>
> *(Scholar P016, MNS Critical Care)*

> *"[I] did my rhinology fellowship, published as first author and coauthor on some [journal] papers".*
>
> *(Scholar P028, MMED ENT)*

*"Since graduation, I have successfully applied to and completed the Fogarty Global Health Fellowship 2020/2021 and was a participant of the Introduction to Research for Young International Academics (IRIYA) 2021 at the Radiological Society of North America (RSNA). I am also currently a fellow under the Multi-morbidity in Uganda Research Capacity Initiative (MURCI) and was awarded a Harvard University Center for AIDS Research (HU-CFAR) award. I also presented work from the master's thesis at the Conference on Retroviral and Opportunistic Infections (CROI) conference 2020 and was awarded an International Investigator Award, and I am currently working as a part-time lecturer at the department of Radiology, Faculty of Medicine of Mbarara University of Science and Technology".*

*(Scholar P021, MMED Radiology)*

*"[I was] promoted to rank as a lecturer, published as a primary author and coauthor, started a surgical simulation course for nose and sinus surgery, [I'm] vice chair faculty curriculum review committee, currently doing a fellowship in curriculum studies and medical education".*

*(Scholar P030, MMED ENT)*

Achievements in research: Notably, more than half of the scholars who reported being in academia were also involved in research. They were at different stages of their research careers, with some being at earlier stages of research career development, while others had already published several times as first or corresponding authors. Through dissemination of their findings in respectable journal articles and conferences, they had enhanced their reputation and career prospects as well. Others were able to train, mentor and supervise undergraduate student research and to support residents in their research projects.

Achievements in innovations and pioneering change: Some of the scholars had introduced new academic programs within their universities and new health services within the hospitals and the communities they served. These innovations were able to bridge some gaps in human resources and healthcare delivery.

*"I implemented the first Ugandan endoscopic sinus surgery dissection course which has run for 3 years now, started a fellowship in curriculum studies and medical education".*

*(Scholar P028, MMED ENT)*

*"I started a surgical simulation course for nose and sinus surgery".*

*(Scholar P030, MMED ENT)*

*"I have successfully spearheaded the formation and operation of UPDF medical aviation services".*

*(Scholar P013, MNS Critical Care)*

Professional advancements: Many scholars reported major successes in their professional careers, such as acquisition of jobs they could not get earlier, job promotions, increased level of confidence in their professional competences and the ability to register and provide specialized care within the communities they served. To further demonstrate their commitment to better clinical care, the scholars were keeping abreast with the current clinical practices through participation in continuous professional development (CPD) trainings organized by various reputable organizations and institutions. One scholar stated it thus:

*"I have been promoted by the health service commission, and I am currently the Head Nurse of the pediatric Intensive Care Unit at Mulago National Referral Hospital".*

*(Scholar P008, MNS Critical Care)*

Additionally, a few of the scholars had been retained by the university to serve initially as assistant lecturers in their departments; some of these had since been promoted to lecturer and other senior positions, including being appointed as heads of departments, an

outcome that effectively positioned them to train, mentor and inspire their junior colleagues to become successful young professionals, as well as to advocate for better training.

> *"I was appointed Assistant Lecturer, to teach histopathology and diagnostic cytology to MLS students".*
>
> *(Scholar P015, MMLS Histopathology)*

*Achievements in leadership:* The scholars had taken on various leadership roles that had enabled them to serve a larger number of people and advance their careers.

> *"I have had the privilege of heading a nursing department and publishing original research papers".*
>
> *(Scholar P031, MNS Critical Care)*

> *"I am vice chair faculty curriculum review committee, currently doing a fellowship in curriculum studies and medical education".*
>
> *(Scholar P030, MMED ENT)*

### 3.2. Scholars' Contributions to Healthcare and Community Development

Based on the qualitative data from the responses to the open-ended question, "State your major contributions to healthcare and community development since obtaining the scholarship", the scholars' major contributions fell into five major themes: clinical care, leadership, research, training and mentorship.

*Specialized clinical care:* The scope of the clinical care services included:

- Specialized nursing care for critically ill patients.
- Supporting perinatal and maternal death audits, leading to reduction in maternal mortality.
- Volunteering in surgical camps in different parts of the country.
- Setting up intensive care units (ICUs), in both private and public hospitals.
- Specialized medical care in MRRH and in community medical camps organized by charity organizations and Rotary.
- Specialized surgical care to the poor and hard to reach areas, e.g., in Karamoja region.

Leadership in healthcare: On top of offering clinical care, some scholars also provided leadership to their teams, including in precarious environments such as war-torn Somalia. It is noteworthy that some of the scholars won awards while still students due to exemplary community leadership.

> *"[I have been] providing quality clinical care to patients and leading the AMISOM medical aviation team in Somalia".*
>
> *(Scholar P013, MNS Critical Care)*

> *"I won the Mijumbi Award 2022 which is awarded the best student in community work and leadership".*
>
> *(Scholar P002, MMED Anesthesia)*

*Health-related research*: The scholars' contributions in the field of health care research covered the entire spectrum of the specializations. They included medical, surgical, nursing, diagnostics and quality improvement. Importantly, many of the scholars had published their works.

> *"I am conducting research in cervical cancer epidemiology, diagnostics and multi-morbidity".*
>
> *(Scholar P015, MMLS Histopathology)*

> *"I am co-investigator on the epidemiology of coronary artery disease (CAD) study. As part of our study, participants undergo CT scan investigations and through this and other study procedures done, we refer any study participants found to have abnormalities to the relevant specialist clinics. I have also published a first-author manuscript based on work from my master's thesis which has also been presented at both local and international conferences".*

*(Scholar P021, MMED Radiology)*

*"I am participating in research development and quality improvement project in my department that improve quality of service delivery".*

*(Scholar P029, MMED Emergency Medicine)*

Training and mentorship: In-service and pre-service trainings and mentorship of students was another major contribution by the scholars. The beneficiaries of these trainings and mentorships by the scholars included undergraduate and postgraduate medical and nursing students, and in-service nurses training in critical care.

*"[I do] clinical work, mentorship and coaching of the various categories of students affiliated to Mulago NRH as well as teaching and supervision of the trainees who undertook a short course in Critical Care Nursing [offered] by [the] Ministry of Health and MUST".*

*(Scholar P008, MNS Critical Care)*

*"I have trained 400 nurses in basic critical care skills who have been spread all over the referral and general hospitals in Uganda".*

*(Scholar P010, MNS Critical Care)*

*"I have been involved in training and assessment of in-service critical care nurses. I also teach direct and advancing (top-up) nursing students at degree concepts in critical care nursing in addition to offering support to Soroti Regional Referral hospital".*

*(Scholar P016, MNS Critical Care)*

*"My teaching and clinical practice was greatly enhanced and so I have been able to participate in training of ICU nurses".*

*(Scholar P019, MNS Critical Care)*

*"Training of ENT students in rhinology skills which are a specialty with very few surgeons in East Africa".*

*(Scholar P028, MMED ENT)*

*"I had the opportunity to train ICU nurses in Kabale and Mbarara in the management of critically ill patients during the COVID 19 pandemic".*

*(Scholar P031, MNS Critical Care)*

## 4. Discussion

This study assessed the GHC scholarship awards, and the employment rates, professional career paths, satisfaction and contributions of the scholarship beneficiaries to health care, as well the scholarship's perceived relevance to the scholars and the community.

### 4.1. GHC Scholarship Awards

The scholarships, of which First Mile were the majority, had mostly been awarded to students of critical care nursing, pediatrics and child health, obstetrics and gynecologists, psychiatry, internal medicine, general surgery and anesthesia. There were few scholars and alumni of emergency medicine, ophthalmology, radiology, ENT, dermatology, pathology and the biomedical sciences. This difference in specialization distribution arises from a strong focus on nursing leadership by the GHC program, and differences in the specialization distribution of the health workforce in various specialties, where established disciplines such as obstetrics and gynecology, general surgery, pediatrics and child health and internal medicine generally attract more applicants compared to non-established specialties such as emergency medicine, pathology and dermatology. This is consistent with a recent study in the same setting [6]. Thus, there is an urgent need for targeted scholarships specifically for programs with inadequate numbers in order to minimize this imbalance.

## 4.2. Employment Rates

This study revealed a high absorption rate of alumni into the workforce. A similar absorption rate was recently reported in the same setting [6], but a lower rate has been reported in a similar setting among a more diverse population of One Health alumni [7,16]. This reflects a growing focus on, and higher demand for, specialized health workers in the country to counteract the negative consequences of task-shifting [17]. In addition, the majority of the respondents in employment were in the public sector, yet the private sector is the largest employer in Uganda. This finding is consistent with general observations of staff mobility from private to public health facilities in the last five years, owing to better packages and job security in the public sector compared to the private sector [18,19]. But this could also indicate possible gaps in the transition from academia to private sector employment through entrepreneurship. Monitoring and evaluation of employment outcomes and job satisfaction have been identified as crucial measures to prepare health professionals for the needs of the markets while at the same time highlighting their own needs [20]. Notably, although research and academia represents a relatively small sector in Uganda, a significant number of alumni were employed there, implying that there might be a need for support for non-academic job transitions [21]. Alternatively, it is possible that these alumni are more likely to respond to surveys of this nature, thus explaining the academia and public sector employment bias observed in this study.

## 4.3. Professional Career Paths and Satisfaction

Our study found that virtually all alumni of the scholarships were registered with their relevant professional bodies, and over 88% of them were employed. Professional registration and licensure are mandatory pre-conditions for health professionals to practice in Uganda where all our respondents were based. The assurance of a job is a strong incentive for someone seeking professional registration and licensure [22]; however, unlike in our study, rampant unlicensed health professional practice has been reported elsewhere in Africa [23] due to lapses in regulatory controls. The scholars expressed that the academic programs and the scholarship were relevant to market needs and their careers, respectively, all of which translated into a high level of career satisfaction.

It is undisputable that graduate training imparts a robust skillset for successful careers in clinical practice, research and academia [21,24,25]. In addition, residency programs have been reported to affect critical aspects of a scholar's career path, including job placement, speed of residency to job transition and engagement in research [26]. Reports have indicated that targeted clinical research programs also boost interest, confidence and skills among students traditionally excluded from science and medicine [27]. Through residency training, the graduates are equipped to become the next generation of global health researchers who will build sustainable partnerships and ensure equity in global health [28]. Career satisfaction of alumni is also essential for maintaining good university–alumni relations, which are vital in mobilizing funding through donations and other initiatives to support the university [29,30].

However, barriers do exist for putting the acquired skillsets effectively into practice. Such barriers include a lack of awareness about career options and limited professional networks outside academia [21]. In our study, although most alumni and students reported being employed and satisfied with their careers, a few were neither employed nor satisfied. While this might be attributed to having only recently graduated, it also implies that there might be gaps in career guidance and support for professional networking that need careful attention to ensure a swift transition from residency program to work. It has been previously reported that helping junior researchers find their niche, seeking alignment between scholarship and work, seeking scholarship outlets and mentorship are valuable strategies in enhancing a smooth transition [31,32].

*4.4. Scholars' Contributions during and after Time on Campus*

Our findings revealed that the scholarship beneficiaries do in fact make valuable contributions to the community during and after their time on campus, through healthcare, training and mentorship, research and innovations and leadership. In essence, the scholars support the core tenets of a modern medical school, particularly in regard to the mandate of social accountability [33], through contribution to the three core roles—teaching, research and service [34]—while on campus. This was further supported by survey findings where 88% of the alumni and 71.4% of the scholars agreed or strongly agreed that they made significant contributions to the medical school and teaching hospital during their campus days. Therefore, having enough residents enrolled in specialty programs and fully supported to focus entirely on their training and clinical work could offer great relief to the healthcare system, particularly in resource-limited settings, which would otherwise have been strained by human resource constraints.

Accordingly, the scholars, through participation in clinical care as residents, help in strengthening the existing mutual collaboration between the university and the teaching hospital—with the medical school providing human resources while the hospital, the training grounds, and these residents in turn receive training and support from both sides. This has a ripple effect on undergraduate medical programs as well. As senior house officers (SHOs), the scholars participate in the clinical training and mentoring of undergraduate students and interns in the hospital. The specialty programs that have so far been supported by the GHC scholarships are among those where the need for continuous mentorship has been strongly expressed by undergraduate students during career sessions [12].

*4.5. Impact of the Scholarships on Nursing Leadership*

It has been suggested that the role of universities is not limited to knowledge production but should include being actors of change by engaging in creative intellectual activities through partnerships [34]. In regard to this, the GHC First Mile program, with its strong focus on empowering nursing leadership to develop and implement innovative models of care, has enabled the MUST Faculty of Medicine to impact nursing care by pioneering the introduction of the Master of Nursing Science (MNS) in critical care nursing (CCN) program in Uganda. Supported by First Mile, the MNS CCN graduates have inspired positive changes in ICU practice, especially with the advent of the COVID-19 pandemic, when intensive care services were in high demand [35,36]. Through this support, the MNS program has now grown beyond CCN to embrace other specializations in pediatric nursing, mental health nursing, community midwifery and reproductive health nursing. The MNS scholars conduct clinical research, publish in reputable journals, deliver clinical care and mentor their juniors through academic instruction.

The success of a limited budget fund such as the GHC scholarship thrives on provision, by the host institution, of a unique, supportive academic environment for the scholar, leveraging funding, being flexible and building strong networks and collaboration between the faculty and the funder [32,37]. Through this approach, the MUST Faculty of Medicine has been able to effectively run the scholarship program with notable impact. This could partly explain the high levels of alumni satisfaction with educational programs, scholarships and careers. A high level of alumni satisfaction has been previously linked with the professionalism of faculty [29], implying that this might be the case for this program, although this study did not evaluate the teaching and learning processes.

*4.6. Study Limitations*

This study relied on the scholars' self-reported data. This may be subject to recall bias, particularly in relation to the on-campus contributions of the scholars who completed their programs much earlier. Our study did not determine how much of the professional success was due specifically to grant funding. However, other studies have shown that financial aid positively affects students' performances and completion in a substantial and statistically robust way [36,38]. It should also be noted that the questionnaire was

designed to make subjective assessment of the impact of the scholarship rather than the postgraduate programs in general. Finally, we also acknowledge the limitation associated with a cross-sectional design, which could lead to response bias. The response rate obtained in this study was low, at 32.5%, which is similar to that of a recent similar survey in the same faculty [6], but still below the average reported in the literature for online surveys (44.1%) [39]. This could be due to survey fatigue in the era of increased online surveys in the advent of the COVID-19 pandemic [40]. However, as noted by [41], this might not be of much significance since there is little evidence correlating response rate and response bias, particularly for alumni surveys. Nevertheless, we have provided an exploratory evaluation of the impacts of a scholarship scheme for the first time during its decade-long implementation.

**5. Conclusions**

This study revealed that the MUST Faculty of Medicine scholarship alumni had valuable impacts on their communities by playing various healthcare-related roles with increased levels of responsibility. The scholars generally perceived that both the scholarship and the educational program were relevant for their career advancement and positive impacts in healthcare at the community level. This reaffirms the value of continuous investment in the specialized training of nurses and medical doctors at a specialized level through scholarships and program development. The strengths and gaps identified in the scholarship program and educational content should act as an impetus to inform relevant improvements and standardization of the scholarship program and educational content by GHC managers and the Faculty of Medicine, respectively.

**Supplementary Materials:** The link to the survey protocol is available here: https://ee.kobotoolbox.org/x/IHY6WWiz accessed on 8 June 2024.

**Author Contributions:** Conceptualization and methodology, J.T., J.N., S.A. and A.K.; data collection, J.T., M.J.N., B.T. and D.A.; data analysis and writing—original draft preparation, J.T. and M.J.N.; writing—review and editing, J.T., J.N., S.A., A.K., M.J.N., B.T. and D.A. All authors have read and agreed to the published version of the manuscript.

**Funding:** This research was funded by Massachusetts General Hospital Center for Global Health's project entitled "First Mile: Powering the Academic Medical Center to Delivery Healthcare in the Community in Uganda", supported through the Wyss Medical Foundation.

**Institutional Review Board Statement:** This study was conducted in accordance with the Declaration of Helsinki, and approved by the Institutional Review Board (or Ethics Committee) of Mbarara University of Science and Technology (MUST REC) (reference number MUST-2023-913 and date of approval 8 August 2023).

**Informed Consent Statement:** Informed consent was obtained from all subjects involved in this study.

**Data Availability Statement:** The dataset used and analyzed during the current study is available from the corresponding author on reasonable request.

**Acknowledgments:** The authors appreciate the assistance from Doreen N. Ngonzi during data collection.

**Conflicts of Interest:** The authors declare no conflicts of interest.

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
