# Peer review of "Impact of Global Health Scholarship Programs in the Faculty of Medicine at Mbarara University of Science and Technology"

_ime, doi:10.3390/ime3030017_

Round 1
Reviewer 1 Report
Comments and Suggestions for Authors
The abstract does not summarize why this is important and how it could be applied elsewhere - it provides the results. Lines 114-137 could be moved to the introduction rather than the methods. This program is an important one for building capacity and encouraging scholarship. It is not clear how individuals get the scholarship. Can anyone apply in the area? Qualitative data including themes and quotes are provided, but no details of the analysis and approach used to identity these themes. Some quotes are used more than once. The discussion could include practical ideas on how this could be used or implemented elsewhere so it has broader relevance.
Reviewer 2 Report
Comments and Suggestions for Authors
General Impression
The authors describe the results of a survey of current scholars and alumni of a global health scholarship at a Ugandan Health Science University. Survey respondents expressed broad satisfaction with the program and listed impressive career accomplishments resulting from their training. The article is well written and appropriately structured. With 720 lines of text, the manuscript is exhausting in both length and detail. However, the impressive volume of the work cannot compensate for the limitations of the study in the areas of controls, objectivity and generalizability. As such, the manuscript reads like an optimistic outcomes report for a funding agency rather than a systematic research study. Publication as a research article cannot be recommended.
Detailed critique
1. Controls
The survey summarizes and details the responses of grant recipients. However, it is not clear how much of their professional success is due to grant funding. No effort has been made to compare scholarship recipients to non-scholars in terms of career outcomes, community engagement and scholarly productivity. If recipients provided a subjective assessment of the impact of the grant (e.g. “the following accomplishments are directly related to the scholarship”), it should be included.
2. Objectivity
The study relies solely on self-reported outcomes of a small sample of scholarship recipients. As the authors acknowledge, the low response rate carries a significant risk of bias as it is not sure how many scholars have not responded because they have left the medical career track all together. It would have been helpful to include objective outcomes data, particularly in areas where data collection is feasible (e.g. number of publications, residency placement).
3. Generalizability
The terms and conditions of the scholarship are not fully described, which limits the interpretation of generalizability. Were recipients required to file annual progress reports to document their accomplishments? Did the scholarship require community service? How long is the duration of the program, and how much of it was covered by the scholarship?
4. Data presentation
Since there are no obvious word or page limits in this journal, the authors present an overly detailed description of the survey results in both narrative and summative form. This defeats the purpose of data analysis, which should present the outcomes of a study in an abbreviated, systematic form. Rather than reporting dozens of free-text comments in unabbreviated from, authors should have used common techniques of qualitative research, such as emerging theme analysis.
Round 2
Reviewer 2 Report
Comments and Suggestions for Authors
Thank you for addressing my comments in detail. The easy-to-fix issues have been addressed by the changes to the organization of the results and the rewrite of the discussion. Addressing issues relating to the design of the study is not practical at this point, but discussing these shortcomings has improved the scientific value of the manuscript.